# Increasing stimulus similarity drives nonmonotonic representational change in hippocampus

**Jeffrey Wammes[1,2]\*, Kenneth A Norman[3,4], Nicholas Turk-Browne[1]**

[1]Department of Psychology, Yale University, New Haven, United States; [2]Department of Psychology, Queen's University, Kingston, Canada; [3]Department of Psychology, Princeton University, Princeton, United States; [4]Princeton Neuroscience Institute, Princeton University, Princeton, United States

**Abstract** Studies of hippocampal learning have obtained seemingly contradictory results, with manipulations that increase coactivation of memories sometimes leading to differentiation of these memories, but sometimes not. These results could potentially be reconciled using the nonmonotonic plasticity hypothesis, which posits that representational change (memories moving apart or together) is a U-shaped function of the coactivation of these memories during learning. Testing this hypothesis requires manipulating coactivation over a wide enough range to reveal the full U-shape. To accomplish this, we used a novel neural network image synthesis procedure to create pairs of stimuli that varied parametrically in their similarity in high-level visual regions that provide input to the hippocampus. Sequences of these pairs were shown to human participants during high-resolution fMRI. As predicted, learning changed the representations of paired images in the dentate gyrus as a U-shaped function of image similarity, with neural differentiation occurring only for moderately similar images.

**\*For correspondence:**
jeffrey.wammes@queensu.ca

## Introduction

Humans constantly learn new facts, encounter new events, and see and hear new things. Successfully managing this incoming information requires accommodating the new with the old, reorganizing memory as we learn from experience. How does learning dynamically shape representations in the hippocampus? Our experiences are encoded in distributed representations (*Johnson et al., 2009*; *Polyn et al., 2005*), spanning populations of neurons that are partially reused across multiple memories, leading to overlap. As we learn, the overlapping neural populations representing different memories in the hippocampus can shift, leading to either *integration*, where memories become more similar to one other, or *differentiation*, where memories become more distinct from one another for reviews, see *Brunec et al., 2020*; *Duncan and Schlichting, 2018*; *Ritvo et al., 2019*.

Whether memories integrate or differentiate depends on whether the synapses that are common across the different memories strengthen or weaken. Traditional Hebbian learning models hold that synaptic connections strengthen when the pre-synaptic neuron repeatedly stimulates the post-synaptic neuron, causing them to fire together (*Buonomano and Merzenich, 1998*; *Caporale and Dan, 2008*; *Feldman, 2009*; *Hebb, 1949*). In other words, coactivation of neurons leads to strengthened connections between these neurons. This logic can scale up to the level of many synapses among entire populations of neurons, comprising distributed representations. A greater degree of coactivation among representations will strengthen shared connections and lead to integration. Consistent with this view, arbitrary pairs of objects integrate in the hippocampus following repeated temporal or spatial co-occurrence (e.g. *Deuker et al., 2016*; *Schapiro et al., 2012*). Moreover, new information that builds a link between two previously disconnected events can lead the representations of

the events to integrate (*Collin et al., 2015*; *Milivojevic et al., 2015*; *Tompary and Davachi, 2017*). In other cases, however, coactivation produces the exact opposite outcome — differentiation. For example, hippocampal representations of two faces with similar associations (*Favila et al., 2016*) and of two navigation events with similar routes (*Chanales et al., 2017*) differentiate as a result of learning. Further complicating matters, some studies have found that the same experimental conditions can lead to integration in some subfields of the hippocampus and differentiation in other subfields (*Dimsdale-Zucker et al., 2018*; *Molitor et al., 2021*; *Schlichting et al., 2015*).

Such findings challenge Hebbian learning as a complete or parsimonious account of hippocampal plasticity. They suggest a more complex relationship between coactivation and representational change than the linear positive relationship predicted by classic Hebbian learning. We recently argued (*Ritvo et al., 2019*) that this complex pattern of data could potentially be explained by the nonmonotonic plasticity hypothesis (NMPH; *Detre et al., 2013*; *Hulbert and Norman, 2015*; *Newman and Norman, 2010*; *Ritvo et al., 2019*), which posits a 'U-shaped' pattern of representational change as a function of the degree to which two memories coactivate (*Figure 1*). According to the NMPH, low levels of coactivation between two memories will lead to no change in their overlap; high levels of coactivation will strengthen mutual connections and lead to integration; and moderate levels of coactivation (where one memory is strongly active and the unique parts of the other memory are only moderately active) will weaken mutual connections and lead to differentiation, thereby reducing competition between the memories for later retrieval attempts (*Hulbert and Norman, 2015*; *Ritvo et al., 2019*; *Wimber et al., 2015*). The NMPH has been put forward as a learning mechanism that applies broadly across tasks in which memories compete, whether they have been linked based on incidental co-occurrence in time or through more intentional associative learning (*Ritvo et al., 2019*). The NMPH can explain findings of differentiation in diverse paradigms (e.g. linking to a shared associate: *Chanales et al., 2017*; *Favila et al., 2016*; *Molitor et al., 2021*; *Schlichting et al., 2015*; retrieval practice: *Hulbert and Norman, 2015*; statistical learning: *Kim et al., 2017*) by positing that these paradigms induced moderate coactivation of competing memories. Likewise, relying on the same parameter of coactivation, the NMPH can explain seemingly contradictory findings showing that shared associates (*Collin et al., 2015*; *Milivojevic et al., 2015*; *Molitor et al., 2021*; *Schlichting et al., 2015*) and co-occurring items (*Schapiro et al., 2012*; *Schapiro et al., 2016*) can lead to integration by positing that — in these cases — the paradigms induced strong coactivation.

Importantly, although the NMPH is compatible with findings of both differentiation and integration across several paradigms with diverse task demands, the explanations above are post hoc and do not provide a principled test of the NMPH's core claim that there is a continuous, U-shaped function relating the level of coactivation to representational change. If there were a way of knowing where on the x-axis of this function an experimental condition was located (note that the U-shaped curve in *Figure 1* has no units), we could make a priori predictions about the learning that should take place, but practically speaking this is impossible: A wide range of neural findings on *metaplasticity* (summarized by *Bear, 2003*) suggest that the transition point on the U-shaped curve between synaptic weakening (leading to differentiation) and synaptic strengthening (leading to integration) can be shifted based on experience. In light of this constraint, *Ritvo et al., 2019* argue that the key to robustly testing the NMPH account of representational change is to obtain samples from the full x-axis of the U-shaped curve and to look for a graded transition where differentiation starts to emerge at higher levels of memory coactivation and then disappears for even higher levels of memory coactivation.

No existing study has demonstrated the full U-shaped pattern for representational change; that is what we set out to do here, using a visual statistical learning paradigm — specifically, we brought about coactivation using temporal co-occurrence between paired items, and we manipulated the degree of coactivation by varying the visual similarity of the items in a pair. *Figure 1* illustrates the NMPH's predictions regarding how pairing two items (A and B) in a visual statistical learning paradigm (such that B reliably follows A) can affect the similarity of the hippocampal representations of A and B. The figure depicts a situation where items A and B have moderate visual similarity, and statistical learning leads to differentiation of their hippocampal representations (because item A's hippocampal representation is moderately activated during the presentation of item B). Crucially, the figure illustrates that there are three factors that influence how strongly the hippocampal representation of item A coactivates with the hippocampal representation of item B during statistical learning:

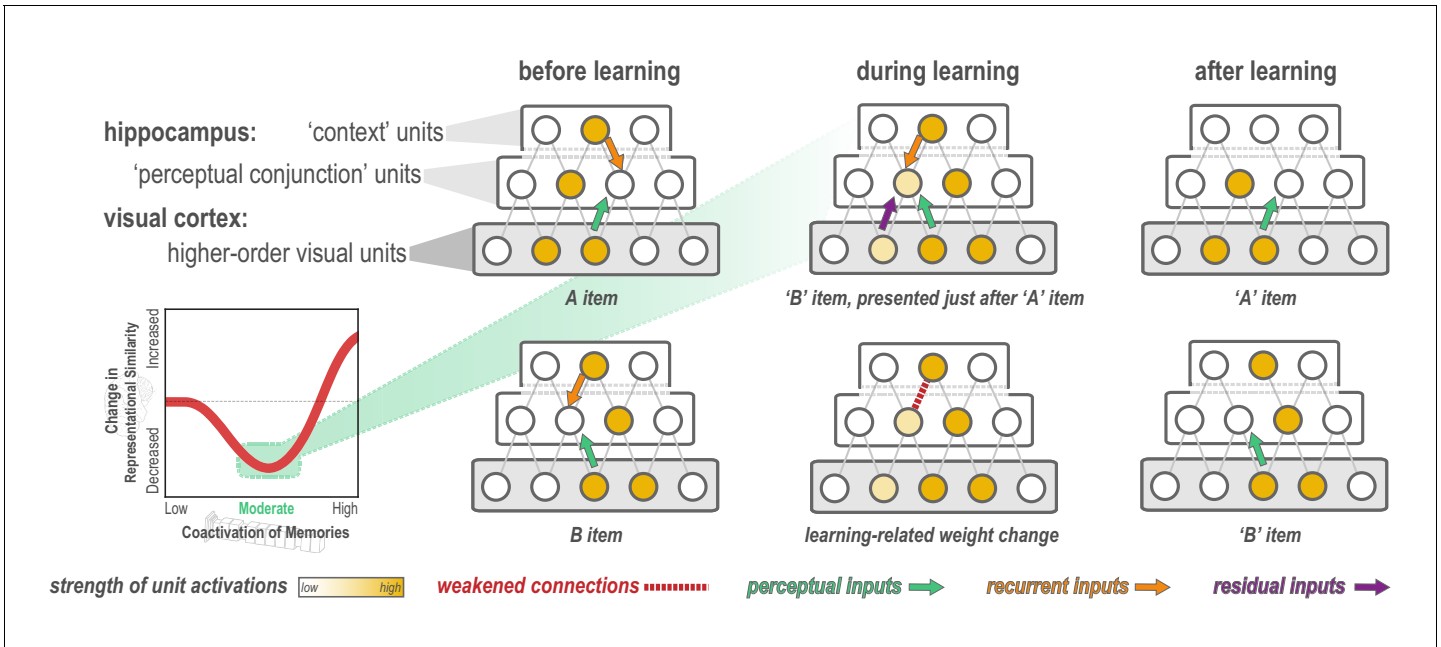

**Figure 1.** Explanation of why moderate levels of visual similarity lead to differentiation. Inset (bottom left) depicts the hypothesized nonmonotonic relationship between coactivation of memories and representational change from pre- to post-learning in the hippocampus. Low coactivation leads to no representational change, moderate coactivation leads to differentiation, and high coactivation leads to integration. Network diagrams show activity patterns in high-level visual cortex and the hippocampus evoked by two stimuli (A and B) with a moderate level of visual similarity that are presented as a 'pair' in a statistical learning procedure (such that B is reliably presented after A). Note that the hippocampus is hierarchically organized into a layer of *perceptual conjunction units* that respond to conjunctions of visual features and a layer of *context units* that respond to other features of the experimental context (*McKenzie et al., 2014*). Before statistical learning (left-hand column), the hippocampal representations of A and B share a context unit (because the items appeared in a highly similar experimental context) but do not share any perceptual conjunction units. The middle column (top) diagram shows network activity during statistical learning, when the B item is presented immediately following an A item; the key consequence of this sequencing is that there is residual activation of A's representation in visual cortex when B is presented. The colored arrows are meant to indicate different sources of input converging on the unique part of each item's hippocampal representation (in the perceptual conjunction layer) when the other item is presented: green = perceptual input from cortex due to shared features (this is proportional to the overlap in the visual cortex representations of these items); orange = recurrent input within the hippocampus; purple = input from residual activation of the unique features of the previously-presented item. The purple input is what is different between the pre-statistical-learning phase (where A is not reliably presented before B) and the statistical learning phase (where A is reliably presented before B). In this example, the orange and green sources of input are not (on their own) sufficient to activate the other item's hippocampal representation during the pre-statistical-learning phase, but the combination of all three sources of input is enough to moderately activate A's hippocampal representation when B is presented during the statistical learning phase. The middle column (bottom) diagram shows the learning that will occur as a result of this moderate activation, according to the NMPH: The connection between the (moderately activated) item-A hippocampal unit and the (strongly activated) hippocampal context unit is weakened (note that this is not the only learning predicted by the NMPH in this scenario, but it is the most relevant learning and hence is highlighted in the diagram). As a result of this weakening, when item A is presented after statistical learning (right-hand column, top), it does not activate the hippocampal context unit, but item B still does (right-hand column, bottom), resulting in an overall decrease in the overlap of the hippocampal representations of A and B from pre-to-post learning.

(1) overlap in the high-level visual cortex representations of items A and B; (2) recurrent input from overlapping features within the hippocampus; and (3) residual activation of item A's representation in visual cortex (because item A was presented immediately before item B). Thus, if we want to parametrically vary the coactivation of the hippocampal representations (to span the full axis of *Figure 1* and test for a full 'U' shape), we need to vary at least one of these three factors. In our study, we chose to focus on the first factor (overlap in visual cortex). Specifically, by controlling the visual similarity of paired items (*Molitor et al., 2021*), we sought to manipulate overlap in visual cortex and (through this) parametrically vary the coactivation of memories in the hippocampus.

To accomplish this goal, we developed a novel approach for synthesizing image pairs using deep neural network (DNN) models of vision. These models provide a link from pictures to rich quantitative descriptions of visual features, which in turn approximate some key principles of how the visual system is organized (e.g. *Cichy et al., 2016*; *Op de Beeck et al., 2008*; *Güçlü and van Gerven,*

*2015*; *Khaligh-Razavi and Kriegeskorte, 2014*; *Kriegeskorte, 2009*; *Kriegeskorte, 2015*; *Kubilius et al., 2016*; *Luo et al., 2016*; *Zeiler and Fergus, 2014*). Most critically, later DNN layers correspond most closely to higher order, object-selective visual areas (*Eickenberg et al., 2017*; *Güçlü and van Gerven, 2015*; *Jozwik et al., 2019*; *Khaligh-Razavi and Kriegeskorte, 2014*), and when neural networks are optimized to match human performance, their higher layers predict neural responses in higher-order visual cortex (*Cadieu et al., 2014*; *Yamins et al., 2014*). We reasoned that

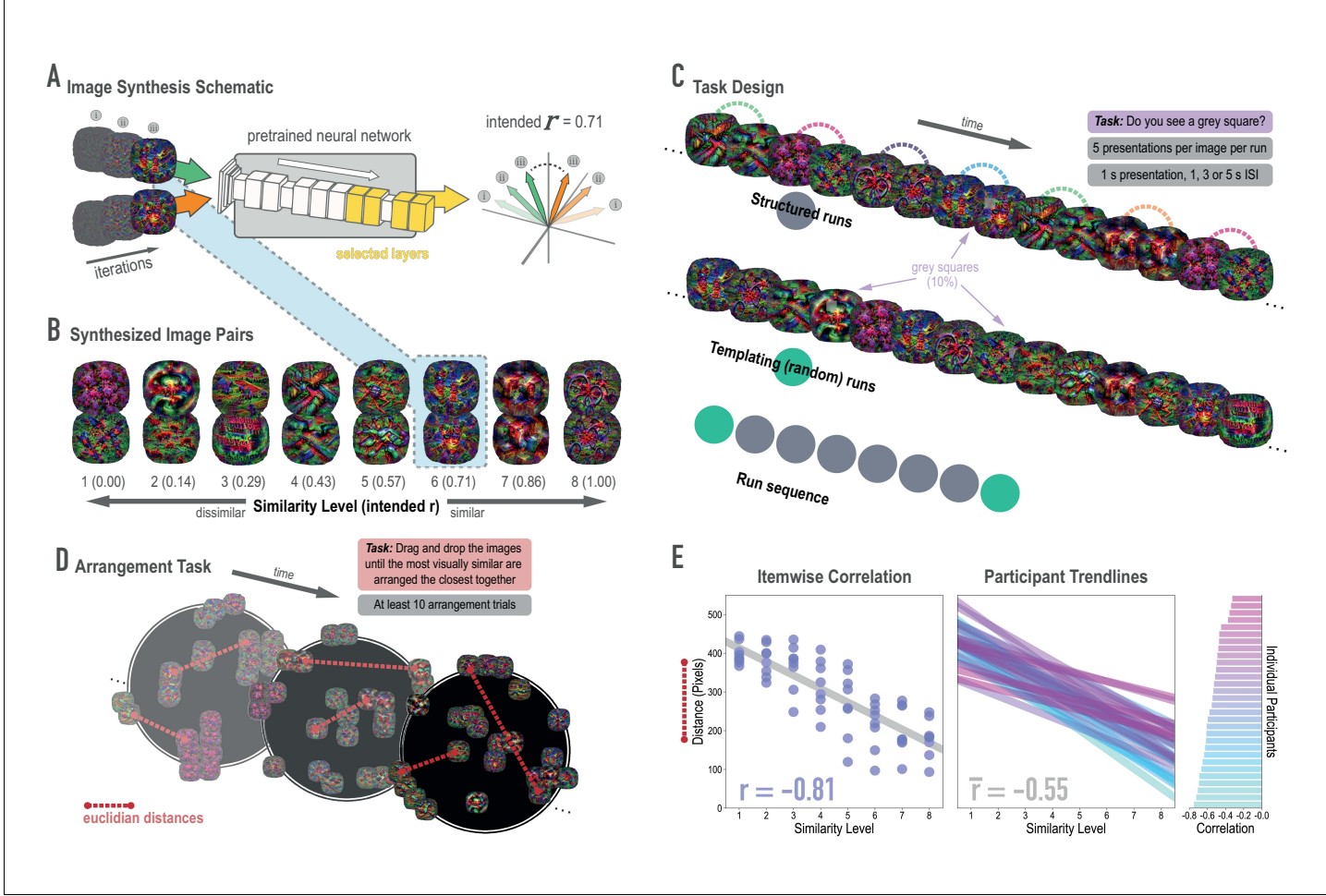

**Figure 2.** Schematic of image synthesis algorithm, fMRI task design, and behavioral validation. (**A**) Our image synthesis algorithm starts with two visual noise arrays that are updated through many iterations (only three are depicted here: i, ii, and iii), until the feature activations from selected neural network layers (shown in yellow) achieve an intended Pearson correlation (r) value. (**B**) The result of our image synthesis algorithm was eight image pairs, that ranged in similarity from completely unrelated (similarity level 1, intended *r* among higher-order features = 0) to almost identical (similarity level 8, intended *r* = 1.00). (**C**) An fMRI experiment was conducted with these images to measure neural similarity and representation change. Participants performed a monitoring task in which they viewed a sequence of images, one at a time, and identified infrequent (10% of trials) gray squares in the image. Unbeknownst to participants, the sequence of images in structured runs contained the pairs (i.e. the first pairmate was *always* followed by the second pairmate); the images in templating runs were pseudo-randomly ordered with no pairs, making it possible to record the neural activity evoked by each image separately. (**D**) A behavioral experiment was conducted to verify that these similarity levels were psychologically meaningful. Participants performed an arrangement task in which they dragged and dropped images in a workspace until the most visually similar images were closest together. From the final arrangements, pairwise Euclidean distances were calculated as a measure of perceived similarity. (**E**) Correlation between model similarity level and distance between images (in pixels) in the arrangement task. On the left, each point represents a pair of images, with distances averaged across participants. In the center, each trendline represents the relationship between similarity level and an individual participants' distances. The rightmost plot shows the magnitude of the correlation for each participant.

The online version of this article includes the following figure supplement(s) for figure 2:

**Figure supplement 1.** Schematic of synthesis algorithm, related to *Figure 2A*.
**Figure supplement 2.** Model validation, related to *Figure 2A,B*.

synthesizing pairs of stimuli that parametrically varied in their feature overlap in the upper layers of a DNN (*Szegedy et al., 2015*) would also parametrically vary their *neural* overlap in the high-level visual regions that provide input to the hippocampus.

Image pairs spanning the range of possible representational overlap values were synthesized according to the procedure shown in *Figure 2A and B* and embedded in a statistical learning paradigm (*Schapiro et al., 2012*). During fMRI, participants were given a pre-learning templating run (where the images were presented in a random order, allowing us to record the neural activity evoked by each image separately), followed by six statistical learning runs (where the images where presented in a structured order, such that the first image in a pair was always followed by the second image), followed by a post-learning templating run (*Figure 2C*). We hypothesized that manipulating the visual similarity of the paired images would allow us to span the x-axis of *Figure 1* and reveal a full U-shaped curve going from no change to differentiation to integration.

We and others have previously hypothesized that nonmonotonic plasticity applies widely throughout the brain (*Ritvo et al., 2019*), including sensory regions (e.g. *Bear, 2003*). In this study, we focused on the hippocampus due to its well-established role in supporting learning effects over relatively short timescales (e.g. *Favila et al., 2016*; *Kim et al., 2017*; *Schapiro et al., 2012*). Importantly, we hypothesized that, even if nonmonotonic plasticity occurs throughout the entire hippocampus, it might be easier to trace out the full predicted U-shape in some hippocampal subfields than in others. As discussed above, our hypothesis is that representational change is determined by the level of coactivation – detecting the U-shape requires sweeping across the full range of coactivation values, and it is particularly important to sample from the low-to-moderate range of coactivation values associated with the differentiation 'dip' in the U-shaped curve (i.e. the leftmost side of the inset in *Figure 1*). Prior work has shown that there is extensive variation in overall activity (sparsity) levels across hippocampal subfields, with CA2/3 and DG showing much sparser codes than CA1 (*Barnes et al., 1990*; *Duncan and Schlichting, 2018*). We hypothesized that regions with sparser levels of overall activity (DG, CA2/3) would show lower overall levels of coactivation and thus do a better job of sampling this differentiation dip, leading to a more robust estimate of the U-shape, compared to regions like CA1 that are less sparse and thus should show higher levels of coactivation (*Ritvo et al., 2019*). Consistent with this idea, human fMRI studies have found that CA1 is relatively biased toward integration and CA2/3/DG are relatively biased toward differentiation (*Dimsdale-Zucker et al., 2018*; *Kim et al., 2017*; *Molitor et al., 2021*). Zooming in on the regions that have shown differentiation in human fMRI (CA2/3/DG), we hypothesized that the U-shape would be most visible in DG, for two reasons: First, DG shows sparser activity than CA3 (*Barnes et al., 1990*; *GoodSmith et al., 2017*; *West et al., 1991*) and thus will do a better job of sampling the left side of the coactivation curve. Second, CA3 is known to show strong attractor dynamics ('pattern completion'; *Guzowski et al., 2004*; *McNaughton and Morris, 1987*; *Rolls and Treves, 1998*) that might make it difficult to observe moderate levels of coactivation. For example, rodent studies have demonstrated that, rather than coactivating representations of different locations, CA3 patterns tend to sharply flip between one pattern and the other (e.g. *Leutgeb et al., 2007*; *Vazdarjanova and Guzowski, 2004*). As discussed below, our hypothesis about DG was borne out in the data: Using synthesized image pairs varying in similarity, we observed the full U-shape (transitioning into and out of differentiation, as a function of similarity) in DG, thereby providing direct evidence that hippocampal plasticity is nonmonotonic.

## Results

### Stimulus synthesis

#### Model validation

Before looking at the effects of statistical learning on hippocampal representations, we wanted to verify that our model-based synthesis approach was effective in creating graded levels of feature similarity in the targeted layers of the network (corresponding to high-level visual cortex): Specifically, our goal was to synthesize images that varied parametrically in their similarity in higher layers while not differing systematically in lower and middle layers of the network. To assess whether we were successful in meeting this goal, we fed the final image pairs (*Figure 2B*) back through the neural network that generated them (GoogLeNet/Inception; *Szegedy et al., 2015*), and computed the

actual feature correlations at the targeted layers. We found that the intended and actual similarity levels of the images (in terms of model features) showed a close correspondence (*Figure 2—figure supplement 2*): In the highest four layers (4D-5B), the intended and actual feature correlations were strongly associated ($r(62)$ = .970, .983, .977, .985, respectively). In the lower and middle layers, feature correlations did not vary across pairs, as intended.

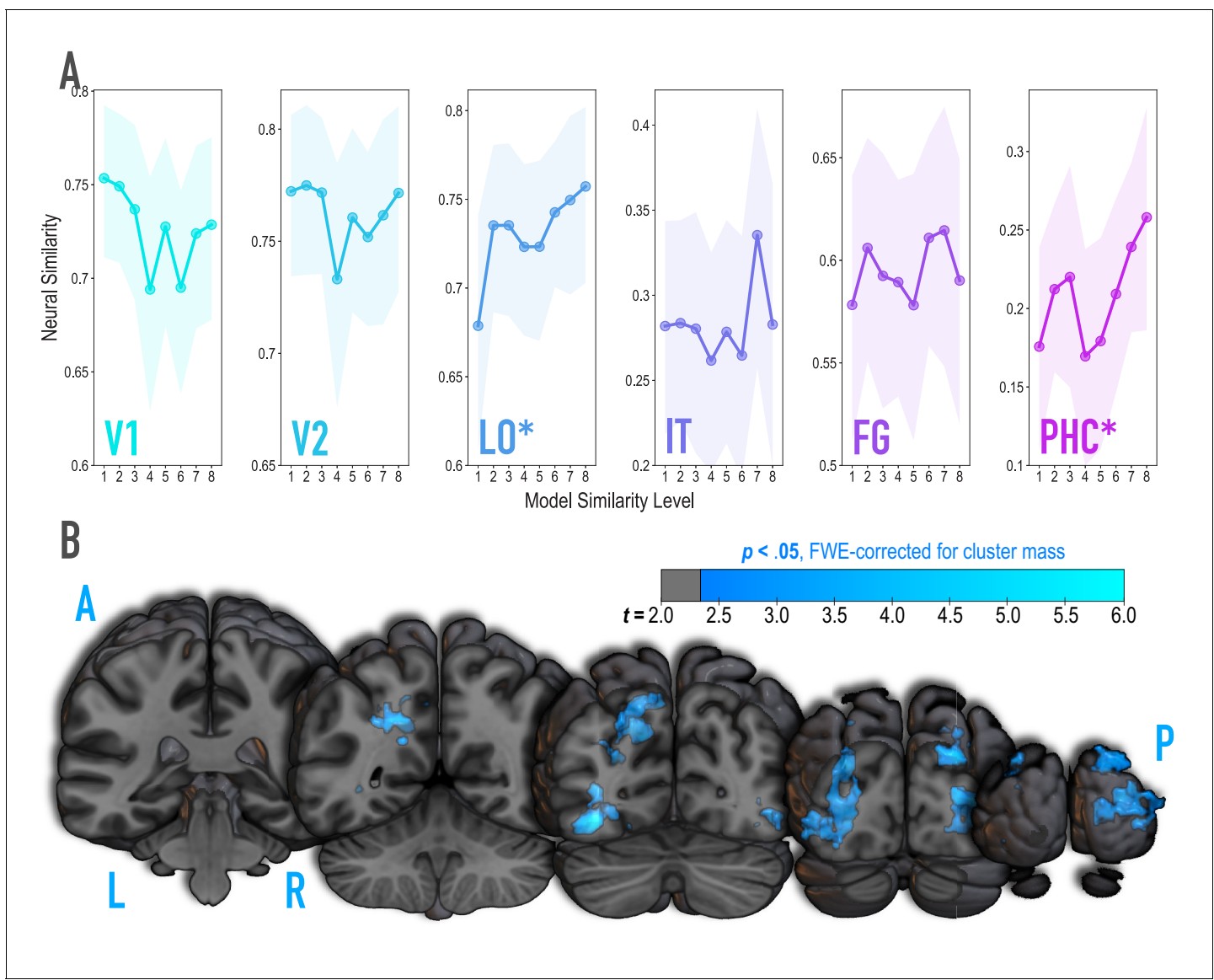

**Figure 3.** Analysis of where in the brain representational similarity tracked model similarity, prior to statistical learning. (**A**) Correlation of voxel activity patterns evoked by pairs of stimuli (before statistical learning) in different brain regions of interest, as a function of model similarity level (i.e. how similar the internal representations of stimuli were in the targeted layers of the model). Neural similarity was reliably positively associated with model similarity level only in LO and PHC. Shaded areas depict bootstrap resampled 95% confidence intervals at each model similarity level. (**B**) Searchlight analysis. Brain images depict coronal slices viewed from a posterior vantage point. Clusters in blue survived correction for family-wise error (FWE) at p < 0.05 using the null distribution of maximum cluster mass. L = left hemisphere, R = right hemisphere, A = anterior, P = posterior.

The online version of this article includes the following figure supplement(s) for figure 3:

**Figure supplement 1.** Similarity tracked model similarity prior to statistical learning, related to *Figure 3*.

**Figure supplement 2.** Analyses of medial temporal cortex subregions, showing neural similarity (prior to statistical learning) as a function of model similarity, related to *Figure 3A*.

## Behavioral validation

Because deep neural networks can be influenced by visual features to which humans are insensitive (*Nguyen et al., 2015*), we also sought to validate that the differences in similarity levels across image pairs were perceptually meaningful to human observers. We employed a behavioral task in which participants (*n* = 30) arranged sets of images (via dragging and dropping in a 2-D workspace; *Figure 2D*), with the instruction to place images that are visually similar close together and images that are visually dissimilar far apart (*Kriegeskorte and Mur, 2012*). Participants completed at least 10 arrangement trials and the distances for each synthesized image pair were averaged across these trials. When further averaged across participants, perceptual distance was strongly negatively associated with the intended model similarity (*r*(62) = −0.813, p < 0.0001; *Figure 2E*). In other words, image pairs at the highest similarity levels were placed closer to one another. In fact, every individual participant's correlation was negative (mean *r* = −0.552, 95% CI = [−0.593–0.512]).

## Neural validation

Because we were synthesizing image pairs based on features from the highest model layers, we hypothesized that model similarity would be associated with representational similarity in high-level visual cortical regions such as lateral occipital (LO) and inferior temporal (IT) cortices. We also explored ventral temporal regions parahippocampal cortex (PHC) and fusiform gyrus (FG), and early visual regions V1 and V2. Based on separate viewing of the 16 synthesized images during the initial templating run (prior to statistical learning), we calculated an image-specific pattern of BOLD activity across voxels in each anatomical ROI. We then correlated these patterns across image pairs as a measure of neural similarity (*Figure 3A*). Model similarity level was positively associated with neural similarity in LO (mean *r* = 0.182, 95% CI = [0.083 0.279], randomization p = 0.007) and PHC (mean *r* = 0.125, 95% CI = [0.022 0.228], p = 0.029). No other region showed a significant positive relationship to model similarity (V2: mean *r* = −0.029, 95% CI = [−0.137 0.077], p = .674; IT: *r* = .070, 95% CI = [−0.057 0.197], p = 0.145; FG: *r* = 0.056, 95% CI = [−0.055 0.170], p = 0.171), including regions of the medial temporal lobe (perirhinal cortex, and entorhinal cortex; *Figure 3—figure supplement 2*); V1 showed a negative relationship (mean *r* = −0.112, 95% CI = [−0.224–0.003], p = .038). The correspondence between the similarity of image pairs in the model and in LO and PHC is consistent with our use of the highest layers of a neural network model for visual object recognition in image synthesis. The fact that this correspondence was observed in LO and PHC but not in earlier visual areas further validates that similarity was based on high-level features.

It is unlikely that any given model layer(s) will map perfectly and exclusively to a single anatomical region. Accordingly, although we targeted higher-order visual cortex (e.g. LO, IT), the layers we manipulated may have influenced representations in other regions, or alternatively, a subset of the voxels within a given anatomical ROI. To explore this possibility, we performed a searchlight analysis (*Figure 3B*) testing where in the brain neural similarity was positively associated with model similarity. This revealed two large clusters of voxels (p < 0.05 corrected): left ventral and dorsal LO extending into posterior FG (3722 voxels; peak *t*-value = 5.60; MNI coordinates of peak = −37.5,−72.0, −10.5; coordinates of center = −26.7,−71.7, 17.4) and right ventral and dorsal LO extending into occipital pole (3107 voxels; peak *t*-value = 4.82; coordinates of peak = 33.0, –88.5, 10.5; coordinates of center = 30.9, –86.5, 10.6). When this analysis was repeated with a reduced sample of the 36 participants who were also included in the subsequent representational change analyses, these clusters no longer emerged as statistically significant at a corrected threshold.

## Representational change

### Hippocampus

We hypothesized that learning-related representational change in the hippocampus, specifically in DG, would follow a nonmonotonic curve. That is, we predicted a cubic function wherein low levels of model similarity would yield no neural change, moderate levels of model similarity would dip toward neural differentiation, and high levels of model similarity would climb back toward neural integration (*Figure 1* inset). We predicted that this nonmonotonic pattern would be observed in the DG, and possibly CA2/3 subfields, given the predisposition of these subfields (especially DG) to sparse representations and pattern separation.

To test this hypothesis, we extracted spatial patterns of voxel activity associated with each image from separate runs that occurred before and after statistical learning (pre- and post-learning templating runs, respectively). In the templating runs, images were presented individually in a completely random order to evaluate how their representations were changed by learning. The response to each image was estimated in every voxel using a GLM. The voxels from each individual's hippocampal subfield ROIs were extracted to form a pattern of activity for each image and subfield. We then calculated the pattern similarity between images in a pair using Pearson correlation, both before and after learning, and subtracted before-learning pattern similarity from after-learning pattern similarity to index the direction and amount of representational change. A separate representational change score was computed for each of the eight model similarity levels.

To test for the U-shaped curve predicted by the NMPH, we fit a theory-constrained cubic model to the series of representational change scores across model similarity levels (*Figure 4A*). Specifically, the leading coefficient was forced to be positive to ensure a dip, followed by a positive inflection — the characteristic shape of the NMPH (*Figure 1* inset). The predictions of this theory-constrained cubic model were reliably associated with representational change in DG ($r = 0.134$, 95% CI = [0.007 0.267], randomization p = 0.022). The fit was not reliable in CA2/3 ($r = 0.082$, 95% CI = [−0.027 0.191], *p* = 0.13), CA1 ($r = 0.116$, 95% CI = [−0.001 0.231], p = 0.10), or the hippocampus as a whole ($r = 0.084$, 95% CI = [−0.018 0.186], p = 0.15). Model fit was also not reliable in other regions of the medial temporal lobe (PHC, perirhinal cortex, and entorhinal cortex; *Figure 4—figure supplement 1*). Interestingly, in an exploratory analysis, we found that the degree of model fit in DG was predicted by the extent to which visual representations in PRC tracked model similarity (see *Figure 4—figure supplement 2*).

We followed up on the observed effect in DG and determined that there was reliable differentiation at model similarity levels 5 ($\Delta r = −0.093$, 95% CI = [−0.177 −0.007], p < 0.0001) and 6 ($\Delta r = −0.090$, 95% CI = [−0.179 −0.004], p = 0.016). This trough in the center of the U-shaped curve was also reliably lower than the peaks preceding it at level 4 ($\Delta r = 0.129$, 95% CI = [0.019 0.243], p = 0.015) and following it at level 8 ($\Delta r = 0.150$, 95% CI = [0.025 .0271], p = 0.005). The curve showed a trend toward positive representational change, suggestive of integration, for model similarity level 8 ($\Delta r = 0.057$, 95% CI = [−0.034 0.147], p = 0.078).

## Whole-brain searchlight

To determine whether nonmonotonic learning effects were specific to the hippocampus, we ran an exploratory searchlight analysis in which we repeated the above cubic model-fitting analysis over the whole brain (*Figure 4B*). This analysis revealed two reliable clusters (p < 0.05 corrected): right hippocampus extending into PHC and FG (832 voxels; peak *t*-value = 4.97; MNI coordinates of peak = 37.5,–7.5, −15.0; coordinates of center = 32.8,–18.7, −17.1) and anterior cingulate, extending into medial prefrontal cortex (1604 voxels; peak *t*-value = 5.43; coordinates of peak = −7.5, 28.5, 21.0; coordinates of center = −5.0, 28.5, 10.2).

## Discussion

We set out to determine how learning shapes representations in the hippocampus and found that the degree of overlap in visual features determined the nature of representational change in DG. The pattern of results was U-shaped: with low or high overlap, object representations did not reliably change with respect to one another, whereas with moderate overlap, they pushed apart from one another following learning. This is consistent with the predictions of the NMPH (*Ritvo et al., 2019*) and related theories (e.g. *Bienenstock et al., 1982*). Although previous studies have reported evidence consistent with the NMPH (e.g. manipulations that boost coactivation of hippocampal representations lead to differentiation; *Chanales et al., 2017*; *Favila et al., 2016*; *Kim et al., 2017*; *Schlichting et al., 2015*), these studies generally compared only two or three conditions and their results can also be explained by competing hypotheses (e.g. a monotonic increase in differentiation with increasing shared activity). Crucially, the present study is the first to span coactivation continuously in order to reveal the full U-shape predicted by the NMPH, whereby differentiation emerges as coactivation grows from low to moderate and dissipates as coactivation continues from moderate to high.

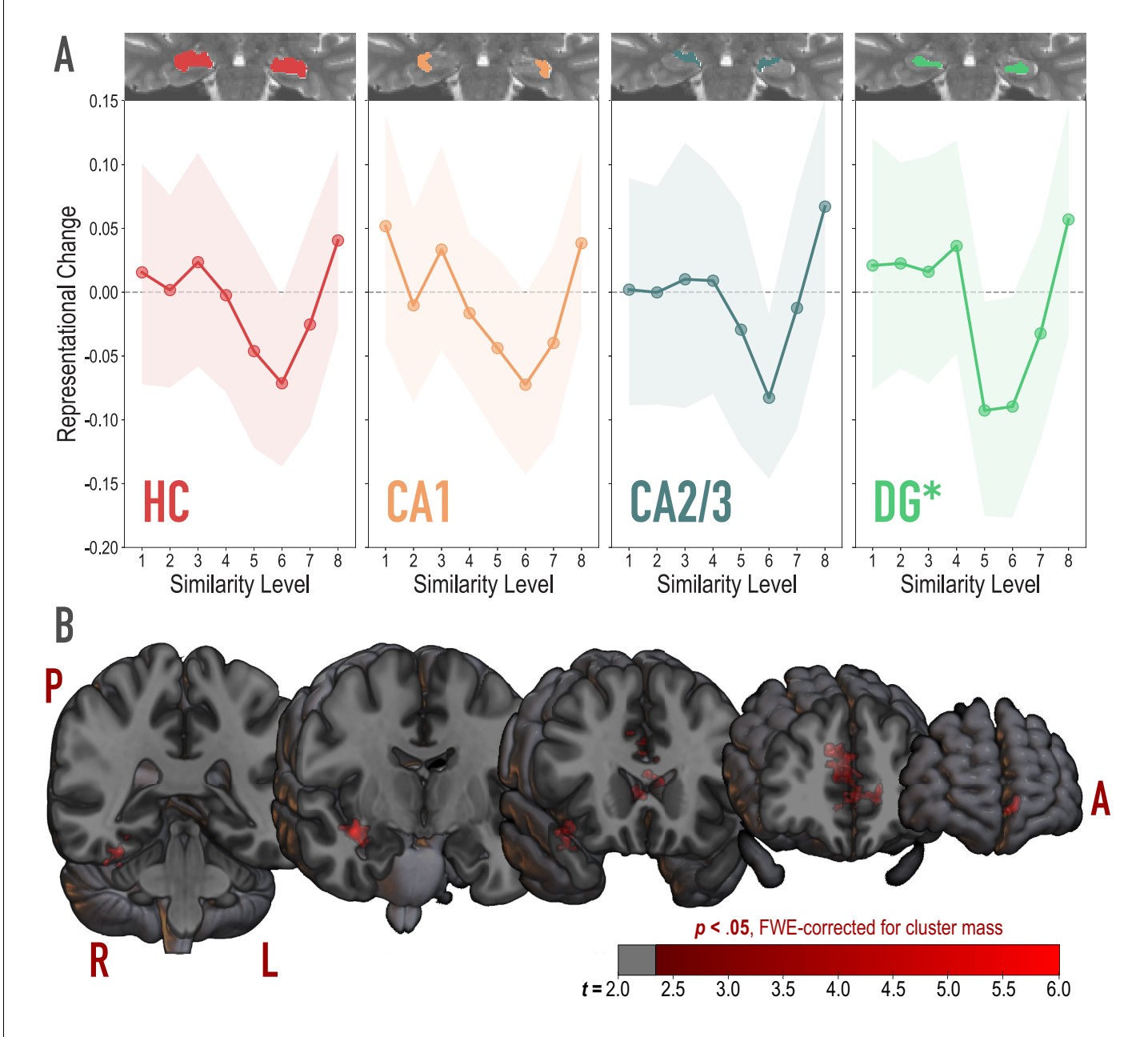

**Figure 4.** Analysis of representational change predicted by the nonmonotonic plasticity hypothesis. (**A**) Difference in correlation of voxel activity patterns between paired images after minus before learning at each model similarity level, in the whole hippocampus (HC) and in hippocampal subfields CA1, CA2/3 and DG. Inset image shows an individual subject mask for the ROI in question, overlaid on their T2-weighted anatomical image. The nonmonotonic plasticity hypothesis reliably predicted representational change in DG. Shaded area depicts bootstrap resampled 95% CIs. (**B**) Searchlight analysis. Brain images depict coronal slices viewed from an anterior vantage point. Clusters in red survived correction for family-wise error (FWE) at p < 0.05 using the null distribution of maximum cluster mass. L = left hemisphere, R = right hemisphere, A = anterior, P = posterior. The online version of this article includes the following figure supplement(s) for figure 4:

**Figure supplement 1.** Analyses of medial temporal cortex subregions, showing representational change as a function of model similarity, related to *Figure 4A*.

**Figure supplement 2.** Exploratory analyses of the association between visual similarity effects in visual regions of interest, and the observed nonmonotonic effect in DG, related to *Figure 4A*.

To measure the impact of the degree of coactivation across a broad range of possible values, we developed a novel method of synthesizing experimental image pairs using DNN models. The intent of this approach was to precisely control the overlap among visual features at one or more layers of the model. In this case, we targeted higher layers of the model to indirectly control representational similarity in higher-order visual regions that provide input to the hippocampus. We found that the imposed visual feature relationships between images influenced human similarity judgments and were associated with parametric changes in neural similarity in higher-order visual cortex (i.e. LO, PHC). This is the first demonstration of the efficacy of stimulus synthesis in manipulating high-level representational similarity in targeted brain regions in humans. These results resonate with recent advances in stimulus synthesis designed to target individual neurons in primates (*Bashivan et al., 2019*; *Ponce et al., 2019*). Although fMRI does not allow for targeting of individual neurons, our findings show that it is feasible to use this method to target distribute representations in different visual cortical regions.

Our approach of manipulating the overlap of visual inputs to the hippocampus, rather than manipulating hippocampal codes directly, was a practical one, based on the fact that we have much better computational models of visual coding than hippocampal coding. Numerous studies have shown that, while hippocampal neurons are indeed 'downstream' from visual cortex, they addition-ally encode complex information from multiple sensory modalities (*Lavenex and Amaral, 2000*), as well as information about reward (*Wimmer and Shohamy, 2012*), social relevance (*Olson et al., 2007*), context (*Turk-Browne et al., 2012*), and time (*Hsieh et al., 2014*; *Schapiro et al., 2012*), to name a few. So, although we controlled visual inputs to the hippocampus, there were many addi-tional non-visual inputs that were free to vary and could play a role in determining the overall rela-tional structure of the representational space. Our work here demonstrates that controlling the visual features alone was sufficient to elicit non-monotonic learning effects, raising the possibility that controlling additional dimensions might yield greater differentiation (or integration). Future work could explore combining models of vision with hippocampal models and attempt to directly target hippocampal representations with image synthesis.

In generating our experimental stimuli, we deliberately avoided the semantic or conceptual infor-mation that comes with meaningful real-world stimuli. This choice was made for several reasons. First, it is highly unlikely that the feature correspondences of even a curated set of real-world stimuli could arrange themselves in a linearly increasing fashion in a targeted model layer, which was a requirement to test our hypotheses. Second, there are known top-down influences on visual repre-sentations (*Gilbert and Li, 2013*), including the integration of conceptual information (*Martin et al., 2018*), which would have undermined our intended visual similarity structure. Last, meaningless pic-tures and shapes are the most commonly used stimuli when studying visual statistical learning (e.g. *Kirkham et al., 2002*; *Luo and Zhao, 2018*; *Schapiro et al., 2012*; *Turk-Browne et al., 2005*), in part to avoid contamination from pre-existing stimulus relationships. Also, although our image pairs were not nameable objects per se, the DNN used to generate them was trained on real-world images (*Deng et al., 2009*), meaning that they were composed from real object features. Neverthe-less, the question remains whether our findings extend to meaningful real-world stimuli. Prior work has shown hippocampal differentiation in experimental conditions involving faces and scenes (*Favila et al., 2016*; *Kim et al., 2014*; *Kim et al., 2017*) as well as complex navigation events (*Chanales et al., 2017*). With this, and assuming one could find a way to impose precise differences in visual overlap as we have here, it seems likely that these effects would generalize. Future work may be able to address this issue more directly using recently developed generative methods (*Son et al., 2020*) or cleverly designed stimuli that capture both conceptual and perceptual similarity (*Martin et al., 2018*).

Importantly, our study allowed us to examine representational change in specific hippocampal subfields. We found that the differentiation 'dip' (creating the U shape) was reliable in DG. This fits with prior studies that found differentiation in a combined CA2/3/DG ROI (*Dimsdale-Zucker et al., 2018*; *Kim et al., 2017*; *Molitor et al., 2021*), and greater sparsity and pattern separation in DG in particular (*Berron et al., 2016*; *GoodSmith et al., 2017*; *Leutgeb et al., 2007*), although note that the U-shaped pattern was trending but not significant in CA2/3 in our study. The clearer effects in DG may suggest that sparse coding (and the resulting low activation levels) is necessary to traverse the full spectrum of coactivation from low to moderate to high that can reveal nonmonotonic changes in representational similarity; regions with less sparsity (and higher baseline activation levels)

may restrict coactivation to the moderate to high range, resulting in a bias toward integration and monotonic increases in representational similarity. Indeed, we had expected that CA1 might show integration effects due to its higher overall levels of activity (*Barnes et al., 1990*), consistent with prior studies emphasizing a role for CA1 in memory integration (*Brunec et al., 2020*; *Dimsdale-Zucker et al., 2018*; *Duncan and Schlichting, 2018*; *Molitor et al., 2021*; *Schlichting et al., 2014*). One speculative possibility is that the hippocampus is affected by feature overlap in earlier stages of visual cortex in addition to later stages (e.g. *Huffman and Stark, 2017*). Our paired stimuli were constructed to have high overlap at the top of the visual hierarchy but low overlap earlier on in the hierarchy; it is possible that allowing stimuli to have higher overlap throughout the visual hierarchy would lead to even greater coactivation in the hippocampus, resulting in integration.

Although we focused above on differences in sparsity when motivating our predictions about sub-field-specific learning effects, there are numerous other factors besides sparsity that could affect coactivation and (through this) modulate learning. For example, the degree of coactivation during statistical learning will be affected by the amount of residual activity of the A item during the B item's presentation in the statistical learning phase. In *Figure 1*, this residual activity is driven by sustained firing in cortex, but this could also be driven by sustained firing in hippocampus; subfields might differ in the degree to which activation of stimulus information is sustained over time (see, e.g. the literature on hippocampal time cells: *Eichenbaum, 2014*; *Howard and Eichenbaum, 2013*), and activation could be influenced by differences in the strength of attractor dynamics within sub-fields (e.g. *Leutgeb et al., 2007*; *Neunuebel and Knierim, 2014*). Also, in *Figure 1*, the learning responsible for differentiation was shown as happening between 'perceptual conjunction' neurons and 'context' neurons in the hippocampus. Subfields may vary in how strongly these item and context features are represented, in the stability/drift of the context representations (*DuBrow et al., 2017*), and in the interconnectivity between item and context features (*Witter et al., 2000*); it is also likely that some of the relevant plasticity between item and context features is happening across, in addition to within, subfields (*Hasselmo and Eichenbaum, 2005*). For these reasons, exploring the predictions of the NMPH in the context of biologically detailed computational models of the hippo-campus (e.g. *Frank et al., 2020*; *Hasselmo and Wyble, 1997*; *Schapiro et al., 2017*) will help to sharpen predictions about what kinds of learning should occur in different parts of the hippocampus.

Although our results are broadly consistent with prior findings that increasing the coactivation of memories can lead to differentiation (*Chanales et al., 2017*; *Favila et al., 2016*; *Kim et al., 2017*; *Schlichting et al., 2015*), they are notably inconsistent with results from *Schapiro et al., 2012*, who reported memory integration for arbitrarily paired images as a result of temporal co-occurrence; pairs in our study with comparable levels of visual similarity (roughly model similarity level 3) showed no evidence of integration. This difference between studies may relate to the fact that the visual sequences in *Schapiro et al., 2012* contained a mix of strong and weak transition probabilities, whereas we used strong transition probabilities exclusively; moreover, our study had a higher base-line of visual feature overlap among pairs. Contextual and task-related factors (*Brunec et al., 2020*), as well as the history of recent activation (*Bear, 2003*), can bias the hippocampus toward integration or differentiation, similar to the remapping based on task context that occurs in rodent hippocampus (*Anderson and Jeffery, 2003*; *Colgin et al., 2008*; *McKenzie et al., 2014*). Speculatively, the over-all higher degree of competition in our task — from stronger transition probabilities and higher baseline similarity — may have biased the hippocampus toward differentiation (*Ritvo et al., 2019*).

Our design had several limitations. Prior work in this area has demonstrated brain-behavior rela-tionships (*Favila et al., 2016*; *Molitor et al., 2021*), so it is clear that changes in representational overlap (i.e. either integration or differentiation) can bear on later behavioral performance. However, in the current work, our behavioral task was intentionally orthogonal to the dimensions of interest (i.e. unrelated to temporal co-occurrence and visual similarity), limiting our ability to draw conclu-sions about potential downstream effects on behavior. We believe that this presents a compelling target for follow-up research. Establishing a behavioral signature of both integration and differentia-tion in the context of nonmonotonic plasticity will not only clarify the brain-behavior relationship, but also allow for investigations in this domain without requiring brain data.

Finally, although analyzing representational overlap in templating runs before and after statistical learning afforded us the ability to quantify pre-to-post changes, our design precluded analysis of the *emergence* of representational change over time. That is, we could not establish whether integration

or differentiation occurred early or late in statistical learning. This is because, during statistical learning runs, the onsets of paired images were almost perfectly correlated, meaning that it was not possible to distinguish the representation of one image from its pairmate. Future work could monitor the time course of representational change, either by interleaving additional templating runs throughout statistical learning (although this could interfere with the statistical learning process), or by exploiting methods with higher temporal resolution where the responses to stimuli presented close in time can more readily be disentangled.

## Conclusion

Overall, these results highlight the complexity of learning rules in the hippocampus, showing that in DG, moderate levels of visual feature similarity lead to differentiation following a statistical learning paradigm, but higher and lower levels of visual similarity do not. From a theoretical perspective, these results provide the strongest evidence to date for the NMPH account of hippocampal plasticity. We expect that a similar U-shaped function relating coactivation and representational change will manifest in paradigms with different task demands and stimuli, but additional work is needed to provide empirical support for this claim about generality. From a methodological perspective, our results provide a proof-of-concept demonstration of how image synthesis, applied to neural network models of specific brain regions, can be used to test how representations in these regions shape learning. As neural network models continue to improve, we expect that this kind of model-based image synthesis will become an increasingly useful tool for studying neuroplasticity.

# Materials and methods

## Participants

For the fMRI study, we recruited 42 healthy young adults participants (18–35 years old, 25 females) with self-reported normal (or corrected to normal) visual acuity and good color vision. All participants provided informed consent to a protocol approved by the Yale IRB and were compensated for their time ($20 per hour). Five participants did not complete the task because of technical errors and/or time constraints, though their data could still be used for the visual templating analyses, as this only required the initial pre-learning templating run. One additional participant's data quality precluded segmentation of hippocampal subfields. As such, our final sample for the learning task was 36 participants, with a total of 41 participants available for the visual templating analyses. See *Figure 3—figure supplement 1* for the outcome of the visual templating analyses in a reduced sample containing only the 36 participants included in the representational change analyses.

For the behavioral validation study, we recruited 30 naive participants through Amazon Mechanical Turk (mTurk). All participants provided informed consent to a protocol approved by the Yale IRB, and were compensated for their time ($6 per hour).

## Stimulus synthesis

Image pairs were generated via a gradient descent optimization using features extracted from *GoogLeNet* a version of *Inception*; (*Szegedy et al., 2015*), a deep neural network (DNN) architecture. This particular instantiation had been pretrained on ImageNet (*Deng et al., 2009*), which contains over one million images of common objects. Accordingly, the learned features reflect information about the real-world features of naturalistic objects. Our approach drew heavily from *Deepdream* (*Mordvintsev et al., 2015*; *Deepdreaming with tensorflow, 2021*), an approach used to visualize the learned features of pretrained neural networks. Deepdream's optimization uses gradient ascent to iteratively update input pixels such that activity in a given unit, layer, or collection of layers is maximized. Different from Deepdream, the core of our approach was controlling the correlation between the features of two images at a given layer *j*, as a means of controlling visual overlap at a targeted level of complexity. We prioritized image optimization over features in a subset of network layers: the early convolutional layers and the output layers of later inception modules (i.e. 12 total layers).

Because we were interested in targeting higher-order visual representations (e.g. in LO), our intention was to produce pairs of images whose higher-layer (top four layers) features were correlated with one another at a specified value, ranging from 0 (not at all similar) to 1 (almost exactly the

same). As such, we produced pairs of images that fell along an axis between two 'endpoints', where the endpoints were pairs of images designed to have a correlation of 0. Each subsequent pair of increasing similarity can be thought of as sampling two new points by stepping inward along the axis from each side. Because it was our aim to have some specificity in the level of representation we were targeting, we sought to fix the feature correlations between all of the pairs in the lower and middle layers (i.e., the bottom eight layers) at 0.25. Altogether, our stimulus synthesis procedure was composed of three phases, described below: (1) endpoint channel selection, (2) image initialization, and (3) correlation tuning (*Figure 2—figure supplement 1*).

The purpose of the endpoint channel selection phase was to select higher-layer feature channels that, when optimized, were maximally different from one another. To do this, we generated an optimized image (*Deepdreaming with tensorflow, 2021*) that maximally expressed each of the 128 feature channels in layer mixed 4E (yielding 128 optimized images). We fed these images back through the network, and extracted the pattern of activation in the top four selected layers for each image. We then computed Pearson correlations among the extracted features and selected the 16 optimized channel images whose activation patterns were least inter-correlated with one another. These 16 channels were formed into eight pairs, which became the endpoint channels for a spectrum of image pairs (*Figure 2—figure supplement 1A*).

During image initialization, the endpoint channels served as the starting point for generating sets of image pairs with linearly increasing visual similarity. For every pair of endpoint channels, eight image pairs were synthesized, varying in intended higher-layer feature correlation from 0 to 1. These are also referred to as model similarity levels 1 through 8. Every AB image pair began with two randomly generated visual noise arrays, and an 'endpoint' was assigned to each — for example, channel 17 to image A and channel 85 to image B. If optimizing image A, channel 17 was always maximally optimized, while the weighting for the optimization of channel 85 depended on the intended correlation. For example, if the intended correlation was 0.14, channel 85 was weighted at 0.14. If optimizing image B for a correlation of 0.14, channel 85 was maximally optimized and channel 17's optimization was weighted at 0.14. These two weighted channel optimizations were added together, and served as the cost function for gradient descent in this phase (*Figure 2—figure supplement 1B*). On each iteration, the gradient of the cost function was computed with respect to the input pixels. In this way, the pixels were updated at each iteration, working toward this weighted image initialization objective. Eight different AB image pairs were optimized for each of the eight pairs of endpoints, for a total of 64 image pairs (128 images in total). After 200 iterations, the resulting images were fed forward into the correlation tuning phase.

The correlation tuning phase more directly and precisely targeted correlation values. On each iteration, the pattern of activations to image A and image B were extracted from every layer of interest, and a correlation was computed between image A's pattern and image B's pattern (*Figure 2A*, *Figure 2—figure supplement 1C*). The cost function for this phase was the squared difference between the current iteration's correlation, and the intended correlation. Our aim was to equate the image pairs in terms of their similarity at lower layers, so the intended correlation for the first eight layers was always 0.25. For the highest four layers, the intended correlation varied from 0 to 1. Similar to the endpoint initialization phase, the gradient of the new cost function was computed with respect to the input pixels, and so the images iteratively stepped toward exhibiting the exact feature correlation properties specified. After 200 iterations, we were left with eight pairs of images, whose correlations (i.e. similarity in higher-order visual features) varied linearly from 0 to 1 (*Figure 2B*, *Figure 2—figure supplement 1C*, right side).

Feature channel optimization tends to favor the expression of high frequency edges. To circumvent this, as in Deepdream (*Mordvintsev et al., 2015*), we utilized Laplacian pyramid regularization to smooth the image and allow low-frequency features and richer color to be expressed. Also, we wanted to ensure that the final feature correlations were not driven by eccentric pixels in the image, and that the feature correlation was reasonably well represented in various parts of the images and at various scales. To accomplish this, we performed the entire 400 iteration optimization procedure three times, magnifying the image by 40% for each volley. We also computed the gradient over a subset of the input pixels, which were selected using a moving window slightly smaller than the image, in the center of the image. As a result of this procedure, pixels toward the outside of the image were not updated as often. We then cropped the images such that pixels that had been iterated over very few times were removed. Although this was intended to avoid feature correlations

driven by the periphery, it had the added benefit of giving the images an irregular ragged edge, rather than a sharp square frame which can make the images appear more homogenous. However, this cropping procedure did remove some pixels that were contributing in part to the assigned correlation value. For this reason, the final inter-image correlations are closely related to but do not *exactly* match the assigned values.

## Model validation

We produced a large set of image pairs using the stimulus synthesis approach detailed above. For each of the eight sets of image endpoints, we generated a pair of images at each of eight model similarity levels, yielding a total of 128 ($8 \times 8 \times 2$) images. Each image was then fed back through the network, resulting in a pattern of activity across units in relevant layers that was correlated with the pattern from its pairmate. We also fed these images through a second commonly used DNN, VGG19 (*Simonyan and Zisserman, 2014*), and applied the same procedure. This was done to verify that the feature overlaps we set out to establish were not specific to one model architecture. In the selected layers of both networks, we computed second-order correlations between the intended and actual image-pair correlations. In the network used to synthesize the image pairs, we averaged these second-order correlations in each of the top four target layers, to provide an estimate of the extent to which the synthesis procedure was effective. Despite varying in similarity in the target layers, the differences in similarity in the lower and middle layers (intended to be uniformly $r$ = 0.25) should be minimal. To confirm this, we calculated differences between the actual correlations across image pairs, as well as the standard deviation of these differences (*Figure 2—figure supplement 2*). Second-order correlations and standard deviations were also computed for each layer in VGG19, the alternate architecture (*Figure 2—figure supplement 2*).

## Behavioral validation

Online participants consented to participate and then were provided with task instructions (see next section). The most critical instruction was as follows: 'It will be your job to drag and drop those images into the arena, and arrange them so that the more visually similar items are placed closer together, and the more dissimilar items are placed farther apart'. After confirming that they understood, participants proceeded to the task. At the start of each trial, they clicked a button labeled ''Start trial''. They were then shown a black screen with a white outlined circle that defined the arena. The circle was surrounded by either 24 (20% of trials) or 26 images (80%), pseudo-randomly selected from the broader set of 128 images (eight pairs from each of 8 sets of endpoints; 64 pairs). Sets were selected such that there were no duplicates, the two images from a given target pair were always presented together, and every image was presented at least twice. Trials were self-paced, but could not last more than 5 min each. Warnings were provided in orange text and then red text when there was 60 and 30 s remaining, respectively. There was no minimum time, except that a trial could not be completed unless every image had been placed. This timing structure ensured that we would get at least 10 trials per subject in no longer than 50 min. On the right side of the arena, there was a button labeled ''Click here when finished''. If this button was clicked prior to placing all of the images, a warning appeared, ''You have not placed all of the images''. If all images had been placed, additional buttons appeared, giving the participant the option to confirm completion (''Are you sure? Click here to confirm.'') or return to sorting (''...or here to go back''). We used the coordinates of the final placement location of each image to compute pairwise Euclidean distances between images (*Figure 2D*). We averaged across trials within participant to get one distance metric for each image pair for each participant. We then computed the Pearson correlation between the model similarity level (1 through 8) defined in the stimulus synthesis procedure and the distance between the images based on each participant's placements. We also averaged Euclidian distances across participants for each of the 64 image pairs, and then computed a Pearson correlation between model similarity level and these group-averaged distances (*Figure 2E*).

## Arrangement task instructions
Instructions (1/5)
Thank you for signing up!

In this experiment, you will be using the mouse to click and drag images around the screen. At the beginning, you will click the 'Start trial' button in the top left corner of your browser window, and a set of images will appear, surrounding the border of a white circular arena. It will be your job to drag and drop those images into the arena, and arrange them so that the more visually similar items are placed closer together, and the more dissimilar items are placed farther apart.

### Instructions (2/5)

You can move each image as many times as you'd like to make sure that the arrangement corresponds to the visual similarity. The images you will be viewing will be abstract in nature, and will not be animals, but the following example should clarify the instructions:

If you had moved images of a wolf, a coyote, and a husky into the arena, you might think that they look quite similar to one another, and therefore place them close together. However, if the next image you pulled in was a second image of a husky, you may need to adjust your previous placements so that the two huskies are closer together than a husky and a wolf.

### Instructions (3/5)

You may find that some clusters of similar items immediately pop out to you. Once you have a set of several of these somewhat similar items grouped together, you might notice more fine-grained differences between them. Please make sure that you take the time to tinker and fine-tune in these situations. The differences within these clusters is just as important.

If two images are exactly the same, they should be placed on top of one another, and if they are ALMOST exactly the same, feel free to overlap one image with the other. By that same logic, if images are totally dissimilar, place them quite far apart, as far as on opposite sides of the arena.

Depending on your screen resolution and zoom settings, the entire circle may not initially be in your field of view. This is okay. Feel free to scroll around and navigate the entire space as you arrange the images. However, it is important that the individual images are large enough that you can make out their details.

### Instructions (4/5)

There will be multiple trials to complete, each with its own set of images. You will have a maximum of 5 min to complete each trial. When there is one minute remaining, an orange message will appear in the center of the circle to inform you of this. You will receive another message, this time in red, when there are 30 s remaining. When time runs out, your arrangement will be saved, and the next trial will be prepared. If you are satisfied with your arrangement before 3 min has elapsed, you can submit it using the 'Click here when finished' button. This will bring up two buttons; one to confirm, and one to go back to sorting. You will not be able to submit your arrangement unless you have placed every image.

When the trial ends, whether it was because you submitted your response, or because time ran out, the screen will be reset, revealing a new set of images and empty arena.

The task will take just under an hour to complete, no matter how quickly or how slowly you complete each trial, so there is no benefit to rushing.

### Instructions (5/5)

We would like to thank you for taking the time to participate in our research. The information that you provide during this task is very valuable to us, and will be extremely helpful in developing our research. With this in mind, we ask that you really pay attention to the details of the images, and complete each trial and arrangement carefully and conscientiously.

[printed in red text] We would also like to remind you that because there is a set time limit, not a set number of trials, there is no benefit to rushing through the trials. In general, we have found that it tends to take 3.5 min at minimum to complete each trial accurately.

## fMRI design

Each functional task run lasted 304.5 s and consisted of viewing a series of synthesized abstract images, one at a time. Images were presented for 1 s each. The first image onset occurred after 6 s, and each subsequent onset was presented after an ISI of 1, 3, or 5 s (40:40:20 ratio). There were

eight pairs, meaning that there were 16 unique images. The order of image presentations was pseudo-randomly assigned in one of two ways. In the first and last (pre- and post-learning) templating runs, the 16 images were presented in a random order for the first 16 trials. For the next 16 trials, the 16 images were presented in a different random order, with the constraint that the same image not be presented twice in a row. This same procedure was repeated until there were 80 total trials (five presentations of each of image). In the six intervening statistical learning runs, image pairs were always presented intact and in the same A to B order. In these runs, the eight pairs were presented in a random order for the first 16 trials. For the next 16 trials, the eight pairs were presented in a different order, with the constraint that the same pair could not be presented twice in a row, and so on. Critically, the images appeared continuously without segmentation cues between pairs, such that participants had to learn transition probabilities in the sequence (i.e. for pair AB, the higher probability of A transitioning to B than B transitioning to any number of other images). One out of every 10 trials was randomly assigned to have a small, partially transparent grey patch overlaid on the image. The participants performed a cover task of pressing a button on a handheld button box when they saw the grey square. This task was designed to encourage participants to maintain attention on the images, but was completely orthogonal to the pair structure.

## Data acquisition

Data were acquired using a 3T Siemens Prisma scanner with a 64-channel head coil at the Yale Magnetic Resonance Research Center. We collected eight functional runs with a with a multiband echo-planar imaging (EPI) sequence (TR = 1500 ms; TE=32.6 ms; voxel size=1.5mm isotropic; FA=71; multiband factor=6), yielding 90 axial slices. Each run contained 203 volumes. For field map correction, two spin-echo field map volumes (TR = 8000; TE=66) were acquired in opposite phase encoding directions. These otherwise matched the parameters of our functional acquisitions. We also collected a T1-weighted magnetization prepared rapid gradient echo (MPRAGE) image (TR = 2300 ms; TE=2.27 ms; voxel size=1 mm isotropic; FA=8; 192 sagittal slices; GRAPPA acceleration factor=3), and a T2-weighted turbo spin-echo (TSE) image (TR = 11390 ms; TE=90 ms; voxel size= $0.44 \times 0.44 \times 1.5$ mm; FA=150; 54 coronal slices; perpendicular to the long axis of the hippocampus; distance factor=20%).

## fMRI preprocessing

For each functional run, preprocessing was performed using the FEAT tool in FSL (*Woolrich et al., 2001*). Data were brain-extracted, corrected for slice timing, high-pass filtered (100 s cutoff), aligned to the middle functional volume of the run using MCFLIRT (*Jenkinson et al., 2002*), and spatially smoothed (3 mm). FSL's topup tool (*Smith et al., 2004*), in conjunction with the two field maps, was used to estimate susceptibility-induced distortions. The output was converted to radians and used to perform fieldmap correction in FEAT. The functional runs were also aligned to both the participants' T1-weighted anatomical image using boundary-based registration, and to MNI standard space with 12 degrees of freedom, using FLIRT (*Jenkinson and Smith, 2001*). Analyses within a single run were conducted in native space. For comparisons across participants, analyses were conducted in standard space.

## Defining regions of interest

For each participant, their T1- and T2-weighted anatomical images were submitted to the automatic segmentation of hippocampal subfields (ASHS) software package (*Yushkevich et al., 2015*), to derive participant-specific medial temporal lobe regions of interest. We used an atlas containing 51 manual segmentations of hippocampal subfields (*Aly and Turk-Browne, 2016a*; *Aly and Turk-Browne, 2016b*). The resulting automated segmentations were used to create masks for the CA1, CA2/3, and dentate gyrus (DG) subfields. For visual ROIs, freesurfer (http://surfer.nmr.mgh.harvard.edu/) was used to create masks for V1, V2, lateral occipital (LO) cortex, fusiform gyrus (FG), parahippocampal cortex (PHC), and inferior temporal (IT) cortex, for each participant.

## General linear model

For the pre- and post-learning templating runs, a regressor was developed for each of the 16 unique synthesized images. This was done by placing a delta function at each image onset and convolving

this time course with the double-gamma hemodynamic response function. We then used these 16 regressors to fit a GLM to the time course of BOLD activity using FSL's FILM tool, correcting for local autocorrelation (*Woolrich et al., 2001*). This yielded parameter estimates for each of the 16 images, which were used for subsequent analyses.

## Stimulus synthesis validation analyses

The pre-learning templating run was analyzed to derive an estimate of the baseline representational similarity among the eight target pairs before any learning had taken place. For ROI analyses, the 16 parameter estimates output by the GLM (one per stimulus) were extracted for each voxel in a given ROI and vectorized to obtain the multivoxel pattern of activity for each stimulus. We then computed the Pearson correlation between the two vectors corresponding to the pairmates in each of the eight target image pairs. This yielded eight representational similarity values, one for each image pair. As established through model-based synthesis, each of the eight image pairs also had a corresponding model similarity level. We computed and Fisher transformed the second-order correlation of neural and model similarity across levels. In other words, this analysis tested whether the pattern of similarity built in to the image pairs through the DNN model corresponded to the representational similarity in a given brain region. We constructed 95% confidence intervals (CIs) for estimates of model-brain correspondence for each ROI by bootstrap resampling of participants 50,000 times. As an additional control, we compared the true group average correlation value to a noise distribution, wherein A and B images were paired randomly 50,000 times, obliterating any systematic similarity relationships among them. We did not, however, constrain the random pairing to exclude the true image pairings. This means that the control is especially conservative because some of the resulting shuffled pairs contained the true image pairings. This analysis was used to ensure that the true effect would not be likely to occur due to random noise. In addition to ROI analyses, we also conducted exploratory searchlight analyses over the whole brain. This involved repeating the same representational similarity analyses but over patterns defined from 125-voxel searchlights (radius = 2) centered on every brain voxel. The resulting whole-brain statistical maps were then Fisher transformed, concatenated, and tested for reliability at the group level using FSL's randomise (*Winkler et al., 2014*). We used a cluster-forming threshold of $t$ = 2.33 ($p$ < .01, one-tailed) in cluster mass, and corrected for multiple comparisons using the null distribution of maximum cluster mass. Clusters that survived at (p < 0.05) were retained.

## Representational change analyses

To test the NMPH, we measured whether learning-related changes in pairmate representational similarity (i.e. changes from pre- to post-statistical-learning) followed a U-shaped, cubic function of model similarity. This involved measuring the degree to which the image pairs at each level of similarity showed integration (i.e. increased representational similarity) or differentiation (i.e. decreased representational similarity). For ROI analyses, the 16 parameter estimates output by the GLM were extracted for each voxel in a given ROI and vectorized. This procedure was performed for both pre- and post-learning templating runs. In each run, Pearson correlations were computed between the vectors for each of the target image pairs, yielding eight representational similarity values. To measure learning-related changes in representational similarity, pre-learning values were subtracted from post-learning values. A positive value on this metric signifies integration, whereas a negative value signifies differentiation. Our predictions were not about any one model similarity level, but rather about a specific nonmonotonic relationship between model similarity and representational change. This hypothesis can be quantified using a cubic model, with the leading coefficient constrained to be positive. To test the efficacy of this model in each MTL ROI, we computed a cross-validated estimate of how well this model predicted the true data. Specifically, we fit the constrained cubic model to the data from all but one held-out participant. We then used the model to predict the held-out participant's data, and computed a correlation between the predicted and actual data. This procedure was repeated such that every participant was held out once, and the resulting correlations were averaged into an estimate of the fit for that ROI. We then constructed 95% confidence intervals (CIs) for estimates of the model fit for each ROI by bootstrap resampling participants 50,000 times. To produce a noise distribution, we randomly re-paired A and B images 50,000 times and repeated the above analysis with our entire sample for each repairing. We compared the true

group average model fit statistic to this noise distribution for each ROI. Lastly, we conducted an exploratory searchlight analysis, repeating the analysis above in 125-voxel searchlights (radius = 2) centered on every brain voxel. As in the model similarity searchlight, all subjects' outputs were then Fisher transformed, concatenated, submitted to randomise, thresholded, and corrected for maximum cluster mass.

## Quantification and statistical analysis

No statistical methods were used to predetermine sample size, but we aimed to collect at least 36 participants for the primary learning analysis. For all analyses, we used bootstrap resampling methods to analyze our results non-parametrically. Details of these analyses, as well as exact results of statistical tests, 95% confidence intervals, and p values with respect to noise distributions, are reported alongside each analysis in our Results. Statistical significance was set at $p < 0.05$ unless otherwise specified.

## Acknowledgements

We thank L Rait for help with recruitment, scheduling, and data collection. This work was supported by NSERC PDF and SSHRC Banting PDF (JDW), NIH R01 MH069456 (KAN and NBT-B), and the Canadian Institute for Advanced Research (NBT-B).

## Additional information

### Funding

| Funder | Grant reference number | Author |
|---|---|---|
| Natural Sciences and Engineering Research Council of Canada | Post-doctoral fellowship | Jeffrey Wammes |
| Social Sciences and Humanities Research Council of Canada | Banting Post-doctoral fellowship | Jeffrey Wammes |
| National Institutes of Health | R01 MH069456 | Kenneth A Norman Nicholas Turk-Browne |
| Canadian Institute for Advanced Research | | Nicholas Turk-Browne |

The funders had no role in study design, data collection and interpretation, or the decision to submit the work for publication.

### Author contributions

Jeffrey Wammes, Conceptualization, Software, Formal analysis, Funding acquisition, Investigation, Visualization, Methodology, Writing - original draft, Project administration, Writing - review and editing; Kenneth A Norman, Nicholas Turk-Browne, Conceptualization, Supervision, Funding acquisition, Methodology, Writing - review and editing

### Author ORCIDs

Jeffrey Wammes (iD) https://orcid.org/0000-0002-8923-5441
Kenneth A Norman (iD) http://orcid.org/0000-0002-5887-9682
Nicholas Turk-Browne (iD) http://orcid.org/0000-0001-7519-3001

### Ethics

Human subjects: All participants provided informed consent, and all research was conducted under a protocol approved by Yale University's Institutional Review Board (IRB Protocol ID: 2000022976).

### Decision letter and Author response

Decision letter https://doi.org/10.7554/eLife.68344.sa1

Author response https://doi.org/10.7554/eLife.68344.sa2

## Additional files

### Supplementary files
• Transparent reporting form

### Data availability
fMRI data are available on Dryad (https://doi.org/10.5061/dryad.t4b8gtj38). Analysis code and synthesized image stimuli are available on GitHub (https://github.com/thelamplab/differint, copy archived at https://archive.softwareheritage.org/swh:1:rev:ed212e5bb9bb-ba8645a4fb3d4152db4030656202). Further information and requests for resources should be directed to and will be fulfilled by the Lead Contact, Jeffrey Wammes (jeffrey.wammes@queensu.ca).

The following dataset was generated:

| Author(s) | Year | Dataset title | Dataset URL | Database and Identifier |
|---|---|---|---|---|
| Wammes J | 2021 | Increasing stimulus similarity drives nonmonotonic representational change in hippocampus | https://doi.org/10.5061/dryad.t4b8gtj38 | Dryad Digital Repository, 10.5061/dryad.t4b8gtj38 |

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
