## [Decision Letter]

**Acceptance summary:**

This paper reports a timely, computationally-inspired fMRI analysis of how hippocampus-dependent memory handles overlap in the timing and visual characteristics of objects we encounter. The elegant experimental approach directly tests the predictions of a theoretical framework by parametrically manipulating visual overlap between associated stimuli. Results showed that within the dentate gyrus of the hippocampus, moderate levels of visual feature similarity led to differentiation following a statistical learning paradigm, but higher and lower levels of visual similarity did not. These findings speak to discrepancies in the field over how the hippocampus responds to similarity in memories and will be of broad interest to memory researchers and computational neuroscientists.

**Decision letter after peer review:**

Thank you for submitting your article "Increasing stimulus similarity drives nonmonotonic representational change in hippocampus" for consideration by *eLife*. Your article has been reviewed by 2 peer reviewers, and the evaluation has been overseen by a Reviewing Editor and Timothy Behrens as the Senior Editor. The following individual involved in review of your submission has agreed to reveal their identity: Thackery Brown (Reviewer #3).

Essential Revisions:

1. Provide a clearer theoretical rationale for the focus on the dentate gyrus given extant rodent work, per the detailed comments provided by Reviewer 3.

2. Address the extent to which these findings are dependent on this particular learning paradigm.

3. Clarify the time course of these effects, in particular as they relate to interpreting the pre- to post-learning results.

4. Describe extent to which these neural patterns meaningfully relate to behaviour.

5. Describe the relationship between the effects in the hippocampus and early visual regions.

*Reviewer #2 (Recommendations for the authors):*

1. Related to point 2 above – it would be helpful to show an analysis of how hippocampal representations change over time. I understand that time is a confound here – representations will drift further from the pre-learning template even with no further learning. However, since all stimuli were presented at all timepoints, adjacent run-to-run similarity values for these pairs could be calculated to establish whether the relative integration/differentiation occurs in a gradual way or as a step function. I leave it up to the authors to decide how to address this, or to provide an explanation for why such an analysis might not be feasible, but this would be helpful in interpreting the pre- to post-learning results.

2. Related to point 3 above – it would be helpful to provide any behavioural measures such as RT that might hint at whether (and how well) the participants learned the links between paired stimuli. Similarly, across the 6 statistical learning runs, did the authors observe RT facilitation for times when the grey square appeared on the second stimulus of a pair (vs. the first one)?

3. Perhaps I'm missing something, but it seems more parsimonious to remove participants who did not have statistical learning data from the templating analyses as well (p. 17). This way, the brain maps and plots are derived from data from the same individuals.

4. Figure 1 is somewhat difficult to follow. The logic is clear, but the lines depicting the information flow between the layers are somewhat difficult to keep in mind – perhaps it would be helpful if the hippocampal and visual layers were displayed in different colours. However, I leave it up to the authors to decide whether to make any changes, this could just be subjective preference. All other figures are exceptionally clear and easy to follow.

*Reviewer #3 (Recommendations for the authors):*

I am very enthusiastic about this work, and did find it quite well-motivated and designed in the broad strokes. I consider my concerns/comments to be "moderate" and am confident they can be addressed.

Specifics on the concerns and my suggestions:

Thinking about the DG findings, the authors present this as being the predicted locus of their non-monotonic representational relationship. But the justification for that is somewhat brief (limited to P.8) and to be honest not immediately intuitive to me. For one thing, in my view of the rodent literature the data might suggest DG favors differentiation for both low and moderate coactivation of memories on such a task, being highly sensitive to small changes. The authors cite the important Leutgeb 2007 work on this, but it's not clear to me that favors DG for the U-shaped pattern observed. The authors might also consider evidence juxtaposing CA1 and CA3 in a similar manner (Vazdarjanova and Guzowski 2004, J Neurosci). That work and the Leutgeb data would seem to favor CA3 having a "thresholded" representational relationship with contextual similarity, whil also suggesting CA1 may have higher discriminability for moderate levels of overlap. One suggestion is that pattern completion processes in the CA3 subcircuit may resist differention from DG up to an extent as contextual similarity drifts.

A related question on the theoretical side is how the network model is being conceptualized for subfields – for example, the subdivisions of the hippocampus can (ought to?) themselves be seen as layers in a neural network. Are the authors envisioning the context and perceptual conjunction cells as layers equally represented in each subfield tested, and that the transformation in that region alone is what differs? I ask because at least in broad terms the inputs to these regions from each other and the entorhinal cortex are different – DG and CA3 are predominantly targeted by entorhinal layer II, while CA1 is largely the target of entorhinal layer III, and it appears "where" and "what" cortical pathways are more segregated in CA1 than CA3 and DG (Witter et al., 2006 Annals of NYAS), and models of hippocampal circuit-level function often view item-context associations in inter-subregional terms (e.g., Hasselmo and Eichenbaum, 2005).

Thinking about the model –

One interesting assumption, if I read correctly, is that residual firing within the hippocampus during statistical learning (in the conjunction units) is "externally driven" – we see a "moderate" repeated activation of a unit in Figure 1 driven by sustained firing in visual cortex from the preceding trial and repeated firing of the same unit in the context layer. To my eye, this would seem to be a high overlap scenario for such a circuit – why does full context repetition and weak input from the cortex not drive strong activity in that item A cell (plasticity has indeed promoted a strong recurrent connection from prior learning between context and the perceptual layer)? Moreover, why do we assume sustained firing in the conjunctive codes in cortex but not those in the hippocampus?

On a related note, it seems allowing context to drift, as in a temporal context model, would help – the context is likely highly similar, but not identical, between adjacent events, which could somewhat attenuate the coactivation of the past event and facilitate the weakened connections with the perceptual conjunction layer.

Perhaps this is lost in the weeds, somewhat for the overall memory – similarity relationships observed, but I think it's an area where more justification and consideration could help tie the fMRI research here to data in rodents and/or motivate continued research into the circuit level of what is being observed.

Overall, I think a bit more on why the model was conceptualized the way it was and how the predictions (or, if more post-hoc – discover) of DG relate to our understanding of neural population connectivity and behavior from animal work would set the study up to promote even more hypothesis testing in this area.

---

## [Author Response]

Essential Revisions:1. Provide a clearer theoretical rationale for the focus on the dentate gyrus given extant rodent work, per the detailed comments provided by Reviewer 3.

We thank the editor for the opportunity to better motivate our focus on the dentate gyrus. We have incorporated new sections in the Introduction to establish why we expected that the DG would be the most likely subfield to exhibit the predicted learning effects. This includes a thorough consideration of the existing rodent work, taking into account the papers helpfully recommended by Reviewer 3. A full accounting of these changes can be found in our response to Reviewer 3’s point #1.

2. Address the extent to which these findings are dependent on this particular learning paradigm.

Our current approach involves a statistical learning task, but non-monotonic plasticity is hypothesized to be a general mechanism that applies across several tasks and learning contexts, and there is evidence from other studies consistent with this idea. We have added sections in the Introduction and Discussion to clarify this point. These changes are described in more detail in our response to Reviewer 2’s point #1.

3. Clarify the time course of these effects, in particular as they relate to interpreting the pre- to post-learning results.

We agree that it would be ideal to be able to understand the time course over which these effects emerge. However, our pre/post design, with intervening statistical learning where transition probabilities were deterministic, precludes such an analysis. We have now explained why this analysis is not possible and included a new passage in the Discussion that highlights this limitation and provides suggestions for future work. These changes are described in detail in our response to Reviewer 2’s point #2.

4. Describe extent to which these neural patterns meaningfully relate to behaviour.

The behavioral cover task we used was orthogonal to the statistical learning and visual similarity manipulations. Grey squares appeared randomly and infrequently and were therefore not predictable. As a result, there is no reason to expect that statistical learning would yield any changes in detection performance. We have now added a section acknowledging this limitation, but pointing toward other work, with similar paradigms, where the same sort of learning resulted in measurable effects in behavior. These changes are described in detail in our response to Reviewer 2’s point #3.

5. Describe the relationship between the effects in the hippocampus and early visual regions.

We thank the editor and reviewer for the fantastic suggestion. In response to this comment, we ran a new analysis where we extracted the linear coefficients for each participant in the visual regions of interest (tracking visual similarity), and the cubic model fit coefficients in the representational change analysis from the hippocampus (tracking non-monotonic plasticity). We found an association between linearity in PRC and non-monotonicity in DG. This analysis is now referenced in the main text and in a figure supplement. These changes are described in detail in our response to Reviewer 2’s point #4.

Reviewer #2 (Recommendations for the authors):1. Related to point 2 above – it would be helpful to show an analysis of how hippocampal representations change over time. I understand that time is a confound here – representations will drift further from the pre-learning template even with no further learning. However, since all stimuli were presented at all timepoints, adjacent run-to-run similarity values for these pairs could be calculated to establish whether the relative integration/differentiation occurs in a gradual way or as a step function. I leave it up to the authors to decide how to address this, or to provide an explanation for why such an analysis might not be feasible, but this would be helpful in interpreting the pre- to post-learning results.

We agree that understanding the emergence of integration or differentiation effects over time would be fantastic, and we appreciate the thoughtful point that in principle it might be possible to compare representations across adjacent runs. However, any signal related to the smaller effect of differential visual representations would be drowned out by the massive confounding effects of temporal auto-correlation. We have summarized these points in our response to your Public Review Point #2, including an added paragraph which both highlights this opportunity for further work, and suggests potential designs or methods that would allow measuring the trajectory of learning.

2. Related to point 3 above – it would be helpful to provide any behavioural measures such as RT that might hint at whether (and how well) the participants learned the links between paired stimuli. Similarly, across the 6 statistical learning runs, did the authors observe RT facilitation for times when the grey square appeared on the second stimulus of a pair (vs. the first one)?

Although statistical learning can facilitate processing of the features of the second item in a pair (e.g., leading to RT facilitation on shape identification: Turk-Browne and Scholl, 2009; Zhao and Luo, 2017; arbitrary category learning: Rogers, Park and Vickery, 2021; and categorical judgments: Turk-Browne, Scholl, Johnson and Chun, 2010; Sherman and Turk- Browne, 2021), the gray square task did not require processing of the features of paired stimuli. Furthermore, the gray squares were inserted randomly in the trial sequence and thus were orthogonal to the pair structure, so there was no way to anticipate when a gray square would appear. As such, there is no reason to expect RT facilitation when the gray square appeared on the second stimulus of a pair compared to the first. Indeed, when we ran this analysis in response to this comment, there was a reduction in RT across runs, but both the main effect of pairmate (first, second) and the interaction of run by pairmate were not significant (*p*s > 0.9). Having said that, we certainly agree about the value of linking representational change to behavior and have now acknowledged this important goal and limitation, as noted above. For what it’s worth, the structure of our task was closely modelled after the structure of Schapiro et al. (2012), in which there was some offline behavioral evidence of statistical learning in terms of familiarity judgments.

3. Perhaps I'm missing something, but it seems more parsimonious to remove participants who did not have statistical learning data from the templating analyses as well (p. 17). This way, the brain maps and plots are derived from data from the same individuals.

Since we had the templating data at our disposal, and the analysis of a given participant’s statistical learning data did not depend on others’ templating data, we opted to include the full sample for the visual analyses. However, we were accidentally inconsistent about this in the previous manuscript: the ROI data (Figure 3A) included only the 36 participants who were included in the learning analysis, whereas the searchlight data (Figure 3B) were reported for all 41 participants. We have now corrected this so both the ROI and searchlight analyses reflect our intention of reporting the full sample of 41 participants. This involved updating the ROI figure and analyses for subpanels 3A and 3B. To address the reviewer’s question, we also ran both analyses with just 36 participants. The pattern of results in the ROI analyses was identical across both approaches (see Figure 3—figure supplement 1). The searchlight analysis also yielded very similar results, with the statistical maps from n = 36 and n = 41 highly correlated across voxels/searchlights (r = 0.928). That said, the clusters reported for n = 41 did not survive correction with n = 36, perhaps reflecting the smaller sample size. We now refer to this in the main text and new supplementary figure.

See Neural validation: **“**When this analysis was repeated with a reduced sample of the 36 participants who were also included in thesubsequent representational change analyses, these clusters no longer emerged as statistically significant at a corrected threshold.”

See Participants: “As such, our final sample for the learning task was 36 participants, with a total of 41 participants available for the visual templating analyses. See Figure 3—figure supplement 1 for visual templating analyses in a reduced sample containing only the 36 participants included in the representational change analyses.”

4. Figure 1 is somewhat difficult to follow. The logic is clear, but the lines depicting the information flow between the layers are somewhat difficult to keep in mind – perhaps it would be helpful if the hippocampal and visual layers were displayed in different colours. However, I leave it up to the authors to decide whether to make any changes, this could just be subjective preference. All other figures are exceptionally clear and easy to follow.

We agree that the hippocampal and visual layers were difficult to discern and that the arrows could have been more prominent. As a result, we have updated the figure based on what the reviewer suggested: the hippocampal and visual layers are now distinguished by a unique background color and the arrows (and labels they correspond to) are larger and color-matched. We thank the reviewer for their suggestions and for the comments about the other figures.

Reviewer #3 (Recommendations for the authors):I am very enthusiastic about this work, and did find it quite well-motivated and designed in the broad strokes. I consider my concerns/comments to be "moderate" and am confident they can be addressed.Specifics on the concerns and my suggestions:1. Thinking about the DG findings, the authors present this as being the predicted locus of their non-monotonic representational relationship. But the justification for that is somewhat brief (limited to P.8) and to be honest not immediately intuitive to me. For one thing, in my view of the rodent literature the data might suggest DG favors differentiation for both low and moderate coactivation of memories on such a task, being highly sensitive to small changes. The authors cite the important Leutgeb 2007 work on this, but it's not clear to me that favors DG for the U-shaped pattern observed. The authors might also consider evidence juxtaposing CA1 and CA3 in a similar manner (Vazdarjanova and Guzowski 2004, J Neurosci). That work and the Leutgeb data would seem to favor CA3 having a "thresholded" representational relationship with contextual similarity, whil also suggesting CA1 may have higher discriminability for moderate levels of overlap. One suggestion is that pattern completion processes in the CA3 subcircuit may resist differention from DG up to an extent as contextual similarity drifts.

We have now clarified our motivation for focusing on DG, drawing upon the very useful references and ideas provided by the reviewer. See Introduction: **“**We and others have previously hypothesized that nonmonotonic plasticity applies widely throughout the brain (Ritvo et al., 2019), including sensory regions (e.g., Bear, 2003). […] For example, rodent studies have demonstrated that, rather than coactivating representations of different locations, CA3 patterns tend to sharply flip between one pattern and the other (e.g., Leutgeb, Leutgeb, Moser, and Moser, 2007; Vazdarjanova and Guzowski, 2004).”

2. A related question on the theoretical side is how the network model is being conceptualized for subfields – for example, the subdivisions of the hippocampus can (ought to?) themselves be seen as layers in a neural network. Are the authors envisioning the context and perceptual conjunction cells as layers equally represented in each subfield tested, and that the transformation in that region alone is what differs? I ask because at least in broad terms the inputs to these regions from each other and the entorhinal cortex are different – DG and CA3 are predominantly targeted by entorhinal layer II, while CA1 is largely the target of entorhinal layer III, and it appears "where" and "what" cortical pathways are more segregated in CA1 than CA3 and DG (Witter et al., 2006 Annals of NYAS), and models of hippocampal circuit-level function often view item-context associations in inter-subregional terms (e.g., Hasselmo and Eichenbaum, 2005).

(see response after point 4)

3. Thinking about the model – One interesting assumption, if I read correctly, is that residual firing within the hippocampus during statistical learning (in the conjunction units) is "externally driven" – we see a "moderate" repeated activation of a unit in Figure 1 driven by sustained firing in visual cortex from the preceding trial and repeated firing of the same unit in the context layer. To my eye, this would seem to be a high overlap scenario for such a circuit – why does full context repetition and weak input from the cortex not drive strong activity in that item A cell (plasticity has indeed promoted a strong recurrent connection from prior learning between context and the perceptual layer)? Moreover, why do we assume sustained firing in the conjunctive codes in cortex but not those in the hippocampus?

(see response after point 4)

4. On a related note, it seems allowing context to drift, as in a temporal context model, would help – the context is likely highly similar, but not identical, between adjacent events, which could somewhat attenuate the coactivation of the past event and facilitate the weakened connections with the perceptual conjunction layer.

We thank the reviewer for the excellent points in comments 2, 3, and 4 above. We completely agree that the factors listed above (the distribution of perceptual conjunction and context neurons and their interconnectivity within and across subfields; the capacity of individual subfields for sustained activation; and temporal drift) can affect learning in our paradigm. We have added a new paragraph to address these points, reprinted from above for convenience.

See Discussion (p. 16, lines 318-336): “Although we focused above on differences in sparsity when motivating our predictions about subfield-specific learning effects, there are numerous other factors besides sparsity that could affect coactivation and (through this) modulate learning. […] For these reasons, exploring the predictions of the NMPH in the context of biologically detailed computational models of the hippocampus (e.g., Schapiro, Turk-Browne, Botvinick, and Norman, 2017; Frank, Montemurro, and Montaldi, 2020; Hasselmo and Wyble, 1997) will help to sharpen predictions about what kinds of learning should occur in different parts of the hippocampus.

5. Perhaps this is lost in the weeds, somewhat for the overall memory – similarity relationships observed, but I think it's an area where more justification and consideration could help tie the fMRI research here to data in rodents and/or motivate continued research into the circuit level of what is being observed. Overall, I think a bit more on why the model was conceptualized the way it was and how the predictions (or, if more post-hoc – discover) of DG relate to our understanding of neural population connectivity and behavior from animal work would set the study up to promote even more hypothesis testing in this area.

We thank the reviewer for their insightful comments; we believe that the resulting edits have substantially strengthened the paper.